# Synthetic Clinical Trial Data while Preserving Subject-Level Privacy

**Mandis Beigi**
Medidata Solutions, a Dassault Systèmes company
New York, NY 10014
Mandis.Beigi@3ds.com

**Afrah Shafquat**
Medidata Solutions, a Dassault Systèmes company
New York, NY 10014
Afrah.Shafquat@3ds.com

**Jason G. Mezey**
Department of Computational Biology, Cornell University
Ithaca, NY 14850
Department of Genetic Medicine, Weill Cornell Medicine
New York, NY 10065
jgm45@cornell.edu

**Jacob W. Aptekar**
Medidata Solutions, a Dassault Systèmes company
New York, NY 10014
Jacob.Aptekar@3ds.com

## Abstract

Clinical trials capture high-quality data for millions of patients each year, yet these data are largely unavailable for research beyond the scope of any individual trial due to a combination of regulatory, intellectual property, and patient privacy barriers. Synthetic clinical trial data that captures the analytical properties of the source data, could provide significant value for research and drug development by making insights widely available while protecting the privacy of the participants. We present a method for generating research-grade synthetic clinical trial data from a real data source. We compared the fidelity and privacy preservation performance of our method to the state-of-the-art deep learning synthesizers and found that our synthesizer had superior performance when applied to clinical trial data as assessed both by established metrics and when considering critical clinical features. We also demonstrate how the privacy settings may be configured to conform to specific privacy policies governing data sharing.

## 1 Introduction

Each year millions of clinical trial participants' data are captured by Electronic Data Capture (EDC) systems [25] to support the development and registration of new therapies. EDC data are regularly analyzed for applications beyond the original clinical trial including for decision-making in pre-clinical and early phase development, development of new technologies to support data capture

NeurIPS 2022 Workshop on Synthetic Data for Empowering ML Research.

and trial monitoring, and post-hoc assessments of trial failures. While the potential research and technology development value of clinical trial data is well appreciated, of the thousands of clinical studies run each year, only a small fraction of the patient level data are broadly available. As with other sensitive data types, data access is limited by regulatory requirement [26], technical protection protocols, the high proprietary value of clinical trial data and the strict privacy requirements required by sponsors, stemming from issues of patient consent and preservation of patient trust. These barriers are significant enough that even with clinical data sharing commitments, policies and protocols, sharing of de-identified patient level clinical trial data remains vanishingly rare.

An alternative solution to providing access to patient level EDC data is the generation of synthetic data, an approach that has been used with some success to share Electronic Health Record (EHR) data. While direct analysis of public and private aggregations of patient level EHR is becoming more common, patient consent and privacy concerns are still a major constraint on general availability of EHR data. For synthetic EHR generation, the major methods currently being deployed have derived from advances in state-of-the art techniques in machine learning and AI, including GANs (Generative Adversarial Networks) [19, 35, 36] and Variational Autoencoders (VAEs) [18], where these methods can demonstrably generate high quality synthetic EHR that preserve critical data characteristics. Deep learning synthetic EHR generation methods are natural comparators because they focus on healthcare data with similar content (e.g., labs, vital signs, clinical observations, medications) and similar privacy interests (e.g., HIPAA, physician-patient confidentiality, insurance underwriting risk). While there are clearly strong parallels between patient level EDC and EHR, there are two key differences which limit the utility of methods generally applied to EHR: differences in data structure and differences in data sharing interests. Regarding differences in structure, EHR are collected sporadically from patient visits to healthcare providers, while EDC data are collected for a precise number of consented participants in controlled experiments designed to test pre-specified hypotheses about the impact of a drug, device or diagnostic on patient outcomes. Consequently, EDC is not only high-dimensional, but less sparse and highly regular (i.e., collected in scheduled or protocolized assessments for each subject under carefully controlled and regulated processes) and generally has smaller total patient numbers, ranging in the tens to hundreds of subjects in a single clinical trial compared to thousands to hundreds of thousands of patients in EHR data from a hospital or health system. While GANs and VAEs are highly flexible, these methods generally require large amounts of training data [10] and perform better for EHR applications where probabilistic associations approximate physical relationships between variables, as opposed to EDC where important structure is rigid and has been imposed by experimental design (e.g., Death Date, if present, must fall strictly between Disease Progression Date and Last Observation Date).

Regarding data-sharing interests, clinical trial sponsors and participants have a very different set of goals and privacy needs than EHR aggregators, patients and providers. First, clinical trial participants explicitly consent to the use of their information for research, subject to an explicit set of policy controls agreed to at the time of the consent which in many cases include consent to secondary analyses or sharing of data. By contrast, most EHR is unconsented for research purposes and is shared based on policies that vary by geography to safeguard privacy interests that have been interpreted by healthcare authorities. Second, EDC data inherits the requirement to preserve the privacy rights of trial subjects [29]. As a result, while EDC data may be shared in qualified circumstances, due to the presence of subject consent and sponsor commitments to when sharing data, the policy requirements on EDC sharing require explicit certification of and control over privacy levels. As the privacy levels attained for GAN and VAE produced synthetic datasets can only be established via post-hoc analysis the lack of pre-specification privacy control makes these deep-learning methods [9, 14] less desirable for producing synthetic clinical trial data. Our method is designed to address these specific issues to produce synthetic clinical trial data. The method is easy-to-implement for generating high-fidelity data from a small sample source and is designed to control privacy for given parameter settings.

To demonstrate these attributes, we used the method to generate high fidelity EDC data from a large repository of clinical trials and compared the results to synthetic data produced by the state-of-the art deep learning methods for which stable implementations are publicly available [23]. For these source data, we show that our method produces synthetic data that is higher fidelity than the established methods by a set of canonical metrics and application-specific metrics, such as survival analysis, where deep learning methods perform significantly worse. We also show that the synthetic data produced achieves the privacy bar of very low risk demanded by EDC patients and data sponsors at individual, attribute and sponsor levels, when considering standard risk metrics and attacker scenarios.

## 2 State-of-the-art Synthesizers

Synthetic data entered the broader public consciousness with advances in deep learning [9, 14] and the advent of generalized adversarial networks (GANs) [15, 31, 32, 33] to create realistic images [10, 19, 34, 36]. Since then, specialized variants have arisen to tackle mixed data types and dependencies beyond those found in image data, which involve modifications to the model architecture, training, and additional processing. The synthesizer medGAN [6, 12], focuses on binary/count data in discrete label patient data through an autoencoder. EMR-(C)WGAN [16] relies on a form of conditional training which requires first encoding patient records into binary vectors for training/input together with batch/layer-normalization.

GretelAI's [21] RNN (Recurrent Neural Network) tokenizes the source dataset and trains on these token sequences. With a validator to ensure that the synthesized data's schema matches the source's schema, it generates synthetic data by prompting the trained Long Short-Term Memory (LSTM) with an initial token whose output sequentially cascades to generate new observations. Though the approach guarantees differential privacy through its modification of gradient updates during training, its tokenization can produce synthetic data whose feature values are invalid or out-of-format strings, requiring manual and careful post-processing and validation. However, apart from CTGAN, these frameworks perform poorly on small datasets. CTGAN [7] can use various data types through a specific encoding of feature values and a re-sampling procedure to account for multimodal data and feature imbalance. The GaussianCopulas model multivariate distributions using copula functions which make the underlying CTGAN model task of learning the data easier. CopulaGAN is a variation of CTGAN which takes advantage of the CDF (Cumulative Distribution Function) based transformation that the GaussianCopulas apply to make the underlying CTGAN model task of learning the data easier.

Overall, aside from their need to train (and possibly pre-train) on large amounts of data and extensive hyper-parameter tuning, these approaches at times fail altogether when training on even moderate amounts of data. Even with sufficient data, training can be difficult in reaching a favorable equilibrium or collapse entirely. In the following sections, we detail how our methodology can train easily and work across a variety of data regimes.

## 3 Proposed synthesis method

For an original source data containing $n$ records, where each record contains $m$ distinct features regarding a subject, with features that can be a mix of categorical and numerical variables such as age, weight, sex, race, treatment, death-flag, etc. these data are first pre-processed by encoding the categorical features via label encoding, including the missing values as a new and distinct encoded value for each feature. The original source data is then encoded using one-hot-encoding and the missing values are imputed using any well-known imputation method. Next the pairwise correlation coefficient is calculated between all pairs of features to determine the highly correlated features, where these features will be co-segregated when generating synthetic data. The data is then embedded into a low-dimensional (e.g., 2 or 3 dimensions) feature space using PCA (Principal Component Analysis) [22] or other embedding approach (e.g., t-SNE (t-Stochastic Neighbor Embedding) [2], UMAP (Uniform Manifold Approximation and Projection) [1]). After embedding the data, the $k$ nearest neighbor algorithm is applied, and for each point one (or more) synthetic data points are simulated by randomly permuting the features of its nearby neighbors within a certain radius/distance. The cluster size $k$ adjusts the levels of fidelity and privacy of the synthesized data, where the smaller the cluster size, the higher the fidelity of the generated data and lower levels of privacy and vice versa. To preserve the privacy of patients with distinctive feature values, outliers defined as points having a distance from their closest neighbor that is larger than the $n^{th}$ percentile of the distances of all the points to their closest neighbor are omitted when selecting nearest neighbors. Finally, a multiplicative Gaussian error with a truncated distribution to the range of the features is added, where this error is centered on the feature value of each simulated record and for discrete features, the values are rounded to the closest integer values. The following parameters are configurable: the features to be co-segregated; the ratio of the number of synthetic subjects to the number of real subjects; the rules for setting the cluster size (e.g., the $k$ nearest neighbors, all neighbors within some distance $\epsilon$, the $k$ nearest neighbors until at least $m$ members of a distinct subclass are included); the value for the

$n^{th}$ percentile to define outliers; and the standard deviation of the truncated Gaussian error. Figure 1 shows a schematic representation of our algorithm.

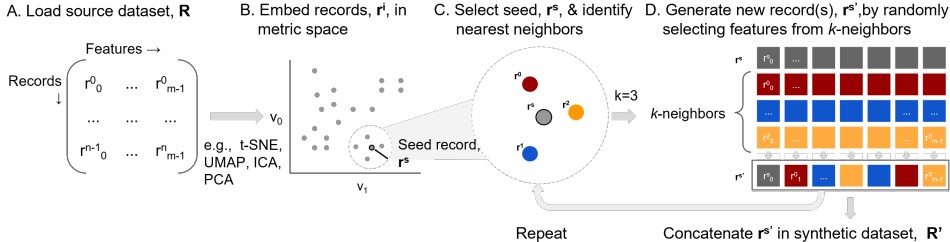

Figure 1: Our proposed method. A. Method is designed to ingest tabular data (i.e., matrix $n$ x $m$). B. Low dimensional embedding by PCA. C. New records are generated from a seed record and its nearest neighbors. The number of neighbors, number of data sources, and number of generations selected for recombination are configurable. D. Attributes are randomly selected from nearest neighbors to produce a synthetic record. In general, the number of possible synthetic children is large relative to the number of parents, $k$. As a result, this generation step is one-to-many and not invertible.

# 4 Synthetic data assessment

## 4.1 Datasets

We evaluated our algorithm's performance in terms of fidelity and privacy on four proprietary clinical trial datasets for different disease indications : (1) Non-Small Cell Lung Cancer consisting of 698 subjects and 171 features, (2) Diffuse Large B-Cell Lymphoma consisting of 1159 subjects and 174 features, (3) Acute Lymphoblastic Leukemia consisting of 4369 subjects and 142 features, and (4) Acute Myeloid Leukemia consisting of 866 subjects and 108 features. These datasets were selected because they each have multiple aspects that make them representative of the type of clinical trial used in drug development and because they have the feature level data usually only available to the owners of clinical trial data. These aspects include (1) each consisted of a clinically homogeneous cohort, (2) the datasets are in ADaM [28] subject level format, allowing assessments of clinically critical factors such as Kaplan Meier curves, (3) each of these datasets has the entirety of relevant data collected in the trial, (4) each includes information on demographics, randomization factors, planned and actual treatment and various subgrouping and population flags.

## 4.2 Fidelity assessment

We used the Synthetic Data Gym (SDGym) [23] from the Synthetic Data Vault (SDV) Project [24] to benchmark out method against four state-of-the-art synthesizers, GaussianCopula, CopulaGAN, CTGAN and TVAE (Tabular Variational AutoEncoder) [7]. The SDV benchmark is a library that offers a set of classical and novel synthetic data generators to use as comparative baselines as well as a large collection of evaluation metrics for cross-validation of the synthetic data against the original data. For numerical features univariate and bivariate tests [44] were used to assess fidelity between the synthesized and source datasets.

The univariate tests included: (1) difference between the synthesized and source datasets in the mean/median values of the feature and the statistical significance of the difference and (2) Kolmogorov-Smirnov test [45] to assess the difference in the distribution of values between the synthetic and source datasets. The bivariate inspections included comparison of the absolute difference in the Pearson correlation coefficient values for all feature pairs between the synthetic and source datasets. For categorical features, fidelity was quantified using the Fisher Exact [45] and Chi-square tests [45] to compute the statistical significance of any differences observed between the synthetic and source datasets. Additionally, we used a multivariate test to quantify the separability of the synthetic dataset from the source dataset using a bag-of-words (BoW) representation (an unsupervised approach). The Silhouette coefficient [45] and discriminative predictive models were trained to distinguish the source data from the synthetic data. Clinical trial specific tests like the Kaplan-Meier curves were also used to compare the differences between synthetic and source datasets

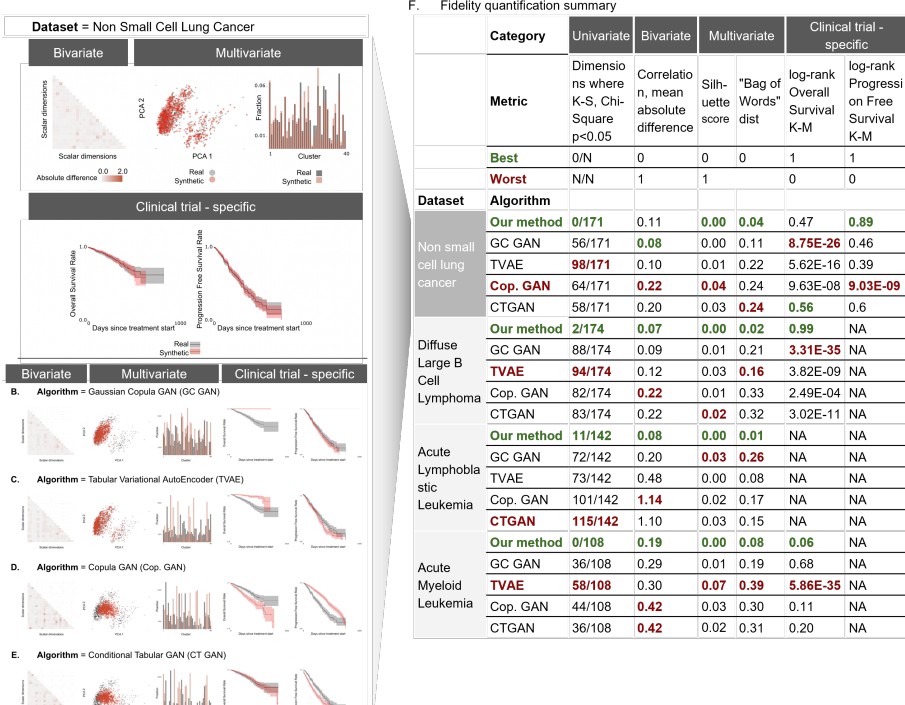

F. Fidelity quantification summary

| Category | Univariate | Bivariate | Multivariate | | Clinical trial - specific | |
|---|---|---|---|---|---|---|
| Metric | Dimensions where K-S, Chi-Square p<0.05 | Correlation, mean absolute difference | Silhouette score | "Bag of Words" dist | log-rank Overall Survival K-M | log-rank Progression on Free Survival K-M |
| **Best** | 0/N | 0 | 0 | 0 | 1 | 1 |
| **Worst** | N/N | 1 | 1 | | 0 | 0 |

| Dataset | Algorithm | | | | | | |
|---|---|---|---|---|---|---|---|
| Non small cell lung cancer | Our method | 0/171 | 0.11 | **0.00** | **0.04** | 0.47 | **0.89** |
| | GC GAN | 56/171 | **0.08** | 0.00 | 0.11 | **8.75E-26** | 0.46 |
| | TVAE | **98/171** | 0.10 | 0.01 | 0.22 | 5.62E-16 | 0.39 |
| | **Cop. GAN** | 64/171 | **0.22** | **0.04** | 0.24 | 9.63E-08 | **9.03E-09** |
| | CTGAN | 58/171 | 0.20 | 0.03 | **0.24** | 0.56 | 0.6 |
| Diffuse Large B Cell Lymphoma | Our method | 2/174 | **0.07** | **0.00** | **0.02** | **0.99** | NA |
| | GC GAN | 88/174 | 0.09 | 0.01 | 0.21 | **3.31E-35** | NA |
| | **TVAE** | **94/174** | 0.12 | 0.03 | **0.16** | 3.82E-09 | NA |
| | Cop. GAN | 82/174 | **0.22** | 0.01 | 0.33 | 2.49E-04 | NA |
| | CTGAN | 83/174 | 0.22 | **0.02** | 0.32 | 3.02E-11 | NA |
| Acute Lymphoblastic Leukemia | Our method | 11/142 | **0.08** | **0.00** | **0.01** | NA | NA |
| | GC GAN | 72/142 | 0.20 | **0.03** | **0.26** | NA | NA |
| | TVAE | 73/142 | 0.48 | 0.00 | 0.08 | NA | NA |
| | Cop. GAN | 101/142 | **1.14** | 0.02 | 0.17 | NA | NA |
| | **CTGAN** | **115/142** | 1.10 | 0.03 | 0.15 | NA | NA |
| Acute Myeloid Leukemia | Our method | 0/108 | **0.19** | **0.00** | **0.08** | 0.06 | NA |
| | GC GAN | 36/108 | 0.29 | 0.01 | 0.19 | 0.68 | NA |
| | **TVAE** | **58/108** | 0.30 | **0.07** | **0.39** | **5.86E-35** | NA |
| | Cop. GAN | 44/108 | **0.42** | 0.03 | 0.30 | 0.11 | NA |
| | CTGAN | 36/108 | **0.42** | 0.02 | 0.31 | 0.20 | NA |

Figure 2: Fidelity results. A-E. The results of bivariate and multivariate tests as well as the overall and progression-free survival Kaplan Meier curves on the non-small cell lung cancer data are shown. Black represents the original source data and red represents the synthesized data using our method, Gaussian Copula GAN, Tabular Variational Encoder, Copula GAN and CTGAN synthesizers (A-E respectively). F. The table shows the quantification of all the tests as illustrated in A-E for all the four datasets

specifically to quantify differences in overall and progression-free survival for the four clinical trial datasets.

Figure 2A shows the results for our method against the four other synthesizers (Figures 2B-2E) on the non-small cell lung cancer dataset. The table shown in Figure 2F summarizes the metrics across all four clinical trial datasets. The heat maps (Figure 2A-2E) show the absolute difference in the Pearson correlation coefficients of the pairwise combinations of all the numerical features between the synthetic and source data. Darker (red) colors show larger differences in correlation coefficient values while the lighter (gray) colors show smaller differences in correlation coefficient values indicating bivariate correlation among most of the feature pairs is maintained. The scatter plots show synthetic subjects (red) and source subjects (black) after being encoded and embedded into a two-dimensional space using Principal Component Analysis (PCA). The histograms show the Bag-of-words representation of the synthetic (red) and source (black) datasets. The Kaplan-Meier curves show the overall survival (OS) and progression-free survival (PFS) for the source (black) and the synthetic (red) subjects in the trial. As shown, our method maintains all the univariate, bivariate, multivariate and clinical trial specific relationships in the non-small cell lung cancer clinical trial dataset (Figures 2A-2E). Our method also showed the best performance among all the leading synthesizers in the benchmark across all four clinical trial datasets (Figure 2F).

## 4.3   Privacy assessment

For the attacker scenario privacy assessments, we prepared different cuts by partitioning the data many times each time varying the size (i.e., number of rows and columns) of the held-out dataset, denoted by $R'$ as illustrated in Figure 3(A-E). The plots in Figure 3(G-I) show the accuracy of training classifiers for feature prediction when a subset of the data is held-out versus held-in. Using RandomForest Regressors and the $R^2$ metric for the accuracy measures, the variable $A_0$ is the y-intercept of the

plotted line represents the accuracy of the machine learning models trained on the held-in dataset $R$ where the accuracy of machine learning models trained on the held-out $R'$ is zero. This coefficient is computed for de-identified dataset and synthesized datasets using our method where the smaller value of $A_0$ represents higher level of privacy and vice versa. Figure 3(H) shows the plots for de-identified data (blue), high fidelity/low privacy versions of the syntehsized data (red) and low fidelity/high privacy synthesized data (green). Figure 3(I) shows a table of all the quantification risks $A_0$ for all the synthesizers on all the four datasets. As expected the de-identified data has the least level of privacy since the patterns, relationships, and exact values in the data are retained and are prone to an attack. From the table in Figure 3(I), our synthesizer and Gaussian Copula have the highest levels of privacy on all the four datasets. However, for the same level of privacy, our method outperforms all the others leading synthesizers in terms of fidelity.

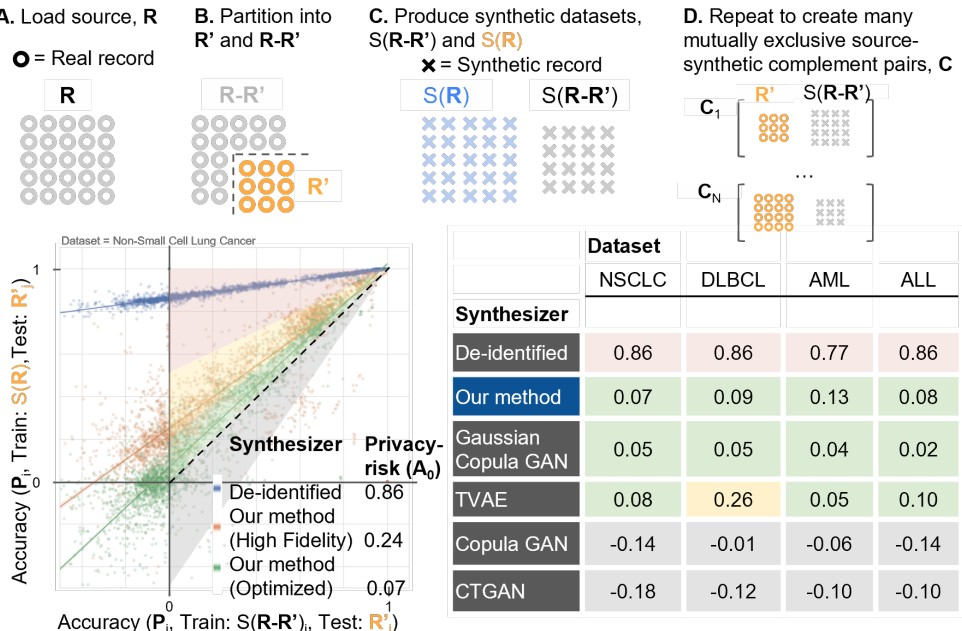

Figure 3: Privacy results. A. Loading of the source dataset $R$. B. Partitioning of the source dataset by removing a subset $R'$. C. Generating synthetic datasets from all of the source dataset R and the remaining dataset after removing $R'$. D. Repeating step C, $N$ times to create mutually exclusive source and synthetic complements. The representative plots for de-identified data (in blue), our method with high fidelity (low privacy) data (in red) and low fidelity (high privacy) data (in green). The table shows the quantification risk $A_0$ for all the synthesizers using all the four datasets.

## 5 Discussion

We note that while our method is designed for the small source data case, for it to produce high fidelity synthetic clinical trial data, there is a sample size limit. Given that an underlying assumption of our method is that any strong conditional relationships for a given measurement are detectable among highly similar individuals, the embedding and neighbor will dictate constraints on the source data sample size. As an example, for PCA embedding, the lower bound on sample size is usually at least five times the number of attributes, and similarly when applying $kNN$ to find neighboring individuals. While this requirement is easily met for the trial data sizes used in our analyses, which are in the order of hundreds to a few thousands, there are many smaller clinical trial cohorts where the application of our method will not be appropriate. For these cases, the synthesis may be performed on aggregated EDC data from multiple trials or when combined with EHR data as long as they have common attributes and are aggregated properly prior to the synthesis.

# 6 Conclusion

The synthesis method we have presented in this paper has three attributes that together are designed to enable the production and sharing of synthetic clinical trial data: (1) the method makes the production of synthetic data easy for the average practitioner, (2) the synthetic data produced has high-fidelity to the source, and (3) a privacy level can be controlled up-front by parameter settings. The method does not require intensive tuning during set-up, variable order (e.g., Sequential Trees) or hyper-parameter optimization (e.g., grid search), or architectural changes in the underlying model structure, which are often required from deep generative frameworks (e.g., Network Architecture Search (NAS)). By comparison, while the value of GANs and related generative machine learning methodologies are clear for EHR data characterized by larger patient sample sizes, these methods have lower performance for smaller sample sizes of clinical trial data when considering both univariate and multivariate performance measures of fidelity.

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
