# OpenReview forum: "Synthetic Clinical Trial Data while Preserving Subject-Level Privacy"
_NeurIPS.cc/2022/Workshop/SyntheticData4ML — Neurips 2022 SyntheticData4ML_

### Official Review · Reviewer_Grh4 · 2022-10-13
**Novel method for generating synthetic tabular data for clinical trials, some open questions about assessment of privacy**

**Rating:** 8
**Confidence:** 5

**Review:**

# Summary

The authors propose a novel method for generating synthetic clinical trial data, addressing the challenges that the small sample size of training data typically bring. The authors also carry out a benchmarking exercise to compare to existing methods.

# Discussion

Pro:
* The proposed method appears to work well in the setting of clinical trials and is appealing due to its simplicity.
* The authors show a comprehensive benchmarking exercise comparing the faithfulness of the generated data to various other methods - this work would already be very useful on its own even outside proposing a new method.

Missing / room for improvement:
* A more detailed exploration / discussion of the parameter space for the model would be helpful (e.g. how much Gaussian noise is added / choice of $k$ for each of their datasets, how were these chosen).
* In the benchmarking exercise it would be good to add the parameters used to train all models. E.g. CTGAN's performance varies depending on the number of epochs used for training (and the out of the box parameter choices are not always optimal).
* The section on the evaluation of privacy is rather short and does not explain how the chosen approach of held-in vs. held-out feature prediction relates to a robust privacy guarantee for the original study participants.
* The figures for the benchmarking exercise are rather small. E.g. the correlation plots are not really readable in their current form - it might be better to just show a numerical summary for these.

# Recommendation

I recommend to accept this paper - models for synthetic data using small training sets such as clinical trial data while protecting participant privacy are a very useful and not very widely studied tool, and the comparative study of different methods is valuable also. Improving the evaluation of privacy (perhaps also across the parameter space of the model) would further improve/strengthen the submission.

---

### Official Review · Reviewer_7Q3F · 2022-10-18
**review of Synthetic Clinical Trial Data while Preserving Subject-Level Privacy**

**Rating:** 6
**Confidence:** 3

**Review:**

Pros:
- Methodology is clear and easy to follow. Appropriate background information present, cites appropriate algorithms and current work.
- There is a rigorous evaluation of the synthetic data generator both against other synthetic data generation methods and also performance against real data across various types of medical data.
- Fits a specific niche of synthesizing clinical trial data (that is already volunteered for research) instead of regular patient data in EHRs.

Cons:
- The key figures comparing the performance of the model are barely legible, they should be larger.
- Citations appear to be out of order.
- I am not well-read in this area of the literature however I feel that more information comparing the algorithms of the other synthetic data generation methods to their new proposed method (a side by side comparison) would be useful to better understand the exact contribution this paper is making to the space.
- Conclusion lacks clear steps on how to apply and improve this method in the future.

---

### Meta-Review · Area_Chair_WvgD · 2022-10-20

**Recommendation:** Accept